

# Stratospheric influence on MLT over mid-latitudes in winter by Fabry-Perot interferometer data

Olga S. Zorkaltseva[1,2], Roman V. Vasilyev[1,2]

[1] Institute of Solar-Terrestrial Physics of Siberian Branch of Russian Academy of Sciences, Irkutsk, 664033, Russia

[2] Irkutsk State University, Irkutsk, 664033, Russia

*Correspondence to:* Olga S. Zorkaltseva (meteorologist-ka@yandex.ru)

**Abstract.** In this paper, we study the response of the mesosphere and lower thermosphere (MLT) to sudden stratospheric warmings (SSWs) and the activity of stationary planetary waves (SPWs). We observe the 557.7-nm optical emission for retrieve the MLT wind, temperature with the Fabry-Perot interferometer (FPI) that has no analogs in Russia. The FPI is

located at the mid latitudes of Eastern Siberia within the Tory Observatory (TOR) at the Institute of Solar-Terrestrial Physics of the Siberian Branch of the Russian Academy of Sciences (ISTP SB RAS, 51.8N, 103.1E). Regular interferometer monitoring started in Dec 2016. Here, we address the temporal variations in the 557.7-nm emission intensity, as well as the variations in wind, temperature, and their variability obtained by using the line parameters measurement during the 2016-2020 winters. Both SSWs and SPWs appear to have equally strong effects in the upper atmosphere. When the 557.7-nm

emission decreases due to some influences from below (SSWs or SPWs), the temperature variation observed by using this line and the temperature itself increase significantly. The zonal wind dispersion does not show significant SPW- and SSW-correlated variations, but the dominant zonal wind reverses during major SSW events the same as the averaged zonal wind at 60N in the stratosphere does without significant delays.

## 1 Introduction

At present, the fact that dynamic processes in various atmospheric layers interact is no doubt. The main mechanism for this interaction is the vertical propagation of atmospheric waves on various temporal and spatial scales. The main role of the atmospheric waves is the energy and momentum transport from the lower atmosphere to the overlying layers, as they propagate. During dissipation in the middle and upper atmosphere, the waves deposit their energy and momentum, thus affecting the thermal balance and circulation of the atmosphere. Hence, the propagation and dissipation of atmospheric

waves is one of the main mechanisms responsible for the energy and dynamic interaction among the lower, middle, and upper atmosphere (Yiğit and Medvedev, 2015)

The mesosphere-lower thermosphere (MLT) is defined as the atmosphere region between about 60 and 110 km in altitude. It constitutes the upper part of what is often referred to as the middle atmosphere (10 to 110 km) (Andrews et al., 1987). Analyzing the observation results showed that the closest relationship between the lower and upper atmospheric layers exists



in winter and early spring (Vincent, 2015). The vertical interaction between the atmospheric layers is especially clearly seen during sudden stratospheric warmings, SSWs (Dowdy et al., 2007, Jacobi et al., 2009). In our paper, we analyze the atmosphere dynamics during the winter season, focusing on SSW events. The goal is not new. There are papers addressing this topic. Below, we summarize the relationship between the background wind and tidal variations in the MLT during stratosphere warming. The main signature of all winter disturbances in the MLT circulation is a significant weakening and

often inversion (east-to-west) of the zonal wind for several days (Danilov et al., 1987). This feature is especially well observed at mid-latitude observatories (Limpasuvan et al., 2016). At polar latitudes, the zonal circulation is less stable, therefore, in some years, the response from the SSW in the dynamics of the MLT may be expressed differently. Most often, the zonal wind reverses westward, and the tides intensify during SSWs (Bhattacharya et al., 2004). Although SSWs are observed in the polar stratosphere, the response in the MLT background winds is recorded at equatorial and tropical

observatories (Sridharan, S. et al., 2012). In (Laskar and Pallamraju, 2014), the authors propose a compelling idea about the existence of a meridional circulation cell in MLT winds during SSW events, which enables the atomic oxygen transport from high to low latitudes. However, we studied a number of papers and could not obtain reliable general conclusions of the meridional wind behavior at mid-latitudes. The MLT meridional wind response to the dynamics of the lower layers differs with different observatories. May be, this is because the general circulation in the upper atmosphere is predominantly zonal

(background meridional wind is smaller).

Like the results of numerous studies show, stratospheric warming has a significant effect on the amplitudes and phases of the MLT tidal oscillations. In (Portnyagin and Sprenger, 1978), the authors divided the tide variations during SSW into two types. In the Type-1 variations, the amplitude of the semidiurnal tide increases significantly and exceeds the amplitude of the diurnal oscillation that is also greater than the usual value during this period. Disturbances of the Type-2 tidal variations are

more complex in their temporal structure and are less common (about 30%). The semidiurnal tide amplitude, in this case, increases shortly (from several hours to a day) and acquires a value close to the diurnal tide amplitude that does not change significantly. The present-day analysis of tides and SSWs confirm this point of view (Pedatella and Liu, 2013).

Monitoring the upper atmosphere over Siberia has been performed since 1976 through various methods. Over 1976–1996, in Eastern Siberia, the MLT wind was monitored by receiving signals from separated radio reception of broadcasting stations in

the long-wavelength range (Vergasova & Kazimirovsky, 2010). From the study (Vergasova & Kazimirovsky, 2010), over Eastern Siberia, the zonal wind at the MLT heights in winter has western directions. Over Badary (51 N, 105 E), during stratospheric warming, as a rule, the zonal wind at the MLT heights was shown to weaken or change direction from west to east. There were cases, when changes in the amplitude and phase of the semidiurnal tide occurred during SSWs. Monitoring the temperature regime in the Siberian region was performed by the hydroxyl emission spectral observations at the Institute

of Solar-Terrestrial Physics of the Siberian Branch of the Russian Academy of Sciences (ISTP SB RAS) Tory (TOR) Observatory (Medvedeva & Ratovsky, 2017; Medvedeva et al., 2014). A significant variation in the OH and O2 emission intensities, a decrease in the atmosphere temperature, and an increase in wave activity were observed during SSWs.





Long-term (covering several years and more) observations of the MLT region temperature and wind are sparse, especially within the Siberian region close to the SSW emergence and evolution. Analysis of such observations is useful to understand

and predict global circulation, and to forecast middle and upper atmospheric models. Our measurements using the Fabry-Perot interferometer enable to simultaneously evaluate the MLT temperature and wind speed. In this paper, we address four winter periods of observations of the upper atmosphere and compare the measurements with the stratosphere dynamics.

## 2 Data and method

We analyze the data from the ISTP SB RAS Fabry-Perot interferometer located at the TOR in the Republic of Buryatia. Fig.

1 shows the map with the instrument location.

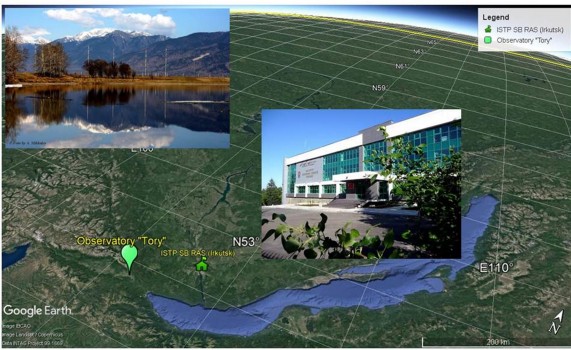

**Figure 1: On the © Google Earth map shows the location of the Fabry-Perot interferometer at the Observatory «Tory» and the Institute of Solar-Terrestrial Physics (ISTP).**

The Fabry-Perot Interferometer (FPI) conducts regular spectrometric observations of the natural airglow lines in the night

atmosphere. Precise spectral analysis enables to observe the Doppler shift of a separate line, which characterizes the movement rate for the corresponding radiating component of the atmosphere along the facility's line-of-sight. The combination of the Doppler shifts obtained in different directions within the medium stationarity time interval enables to reconstruct the full vector of the wind horizontal velocity, whereas the line broadening provides us with the information on the temperature (Vasilyev et al., 2017). In this paper, we address the data on the behavior of the wind speed zonal component

and the temperature obtained by using the 557.7-nm line emission originating at about 90-100 km over the Earth surface. The FPI is an optical instrument, therefore, measurements are possible only in the dark, on moonless nights, when there are no clouds within the FPI field of view. Due to this, the data have periodic (daily, lunar) and aperiodic (cloudiness, technical failures) gaps. In this paper, we analyze the night-averaged values for the 557.7-nm emission intensity (I), temperature (T), and zonal wind speed (U), as well as the standard deviations of those values during each night.





To study the stratosphere dynamics, we used the ECMWF Era5 climate archive (Hersbach et al., 2020). As per the SSW criteria established by the WMO, we address the parameters, such as the zonal average air temperature along 80N and zonal average values of the wind zonal component along 60N at the 10hPa height on a 2.5x2.5 deg grid. We also studied the dynamics of stationary planetary waves with zonal wavenumbers 1 (SPW1) and 2 (SPW2). We addressed all the characteristics at the 1 hPa and 10 hPa heights.

## 3. Results and discussion

### 3.1 Sudden stratospheric warmings

Fig. 2 shows the daily zonal mean zonal wind at 60N (blue) and the temperature at 80N (red) obtained from the ERA5 reanalysis dataset for 1 Oct 2016 through 31 Mar 2017 at the 10 hPa height (solid) and at 1 hPa (dotted). We see that two stratospheric warmings were observed with a peak on Feb 1 (251 K) and Feb 27 (251 K), SSW1 and SSW2, respectively,

marked by dotted vertical lines in the figure. The grey rectangles show the SSW duration. As a criterion for the warming onset, we accepted a day with a sharp temperature increase (more than 10 deg per day). We accepted the sharp (about 2 deg per day) temperature decrease end as the SSW end. SSW1 started on Jan 20 and ended on Feb 12. SSW2 started on Feb 23 and ended on Mar 5. As per the World Meteorological Organization (WMO) standard criteria, observed were two minor warming events during the 2017 winter. Note that the warming at the 1 hPa height was significant, and the zonal circulation

reversed. This may be important to analyze vertical interactions that we address below.

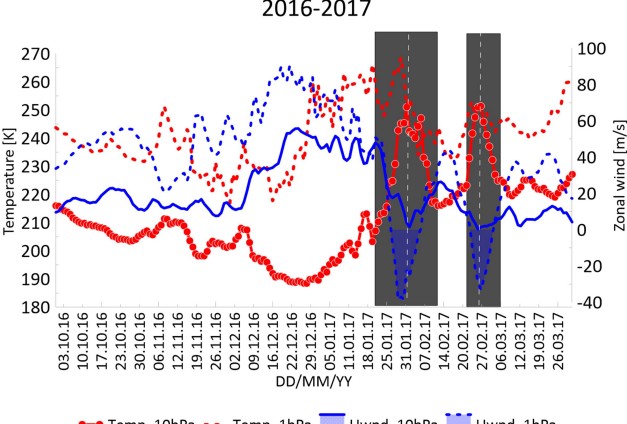

**Figure 2: Zonal mean of the zonal wind at 10 hPa (blue solid line) and 1 hPa (blue dotted line), and zonal mean temperature at 10 hPa (red solid line) and 1 hPa (red dotted line) obtained from ERA5 reanalysis dataset Oct 2016 through Mar 2017.**

In the introduction, we discussed that waves, including planetary waves, are the cause for vertical interaction in the

atmosphere. Periods of increase in the planetary wave amplitude in the stratosphere are not always accompanied by the SSW





evolution. Therefore, we address (and mark on the plots) the periods of planetary wave amplitude increase without SSW, and compare them with the MLT dynamics in the next section. In the figures, we mark the SPW1 amplitude increase above the average value for each winter with a light grey rectangle. Fig. 3 shows that, in early Nov 2016, there was a significant SPW1 amplitude increase that persisted for about a month.

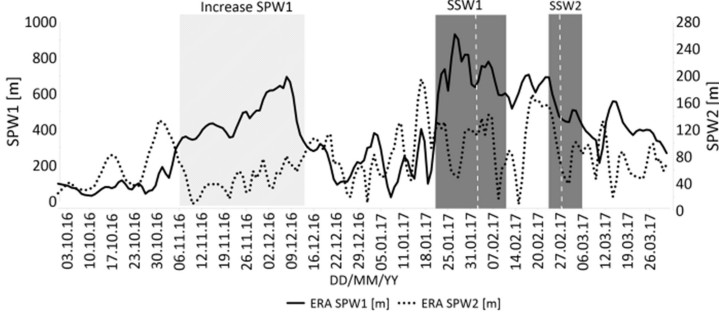


**Figure 3: Amplitude of stationary planetary wave 1 at 10 hPa (solid line), and SPW2 amplitude at 10 hPa (dotted line) Oct 2016 through Mar 2017.**

The SSW spatial structure may also be important for the upper atmosphere response. We analyzed temperature maps during warmings. As an example, we give a temperature map on the SSW maximum day (Fig. 4) at 10hPa. In the 2016-2017 winter, 115 both SSW cases evolved in the Eastern Hemisphere, the warming center was located near the FPI location.

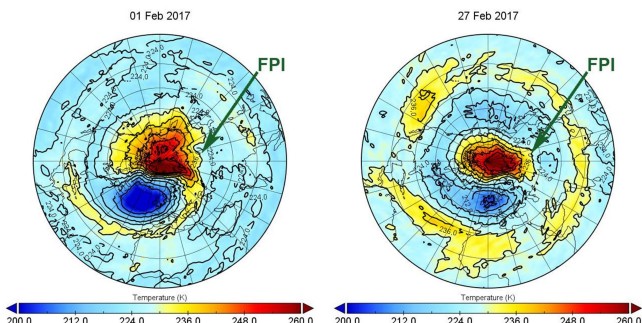

**Figure 4: Distribution of temperature at 10 hPa in stereographic projection at the SSW maximum. Green arrow shows the FPI location.**

In the 2017-2018 winter, there was one SSW case that started on Feb 10 and ended on Mar 2. The maximal temperature was 246K by Feb 18. In Fig. 5, we can see that the warming was major, the zonal wind inversion was observed at 10 hPa and 1 hPa. The warming predominantly evolved in the Western Hemisphere over America, and the FPI was at the SSW periphery





(Fig. 7). Note that the 1-hPa temperature decreased during the SSW. Before the SSW onset, two SPW1 increases were observed in the stratosphere. An SPW1 increase was noted on Dec 2017, as well as from mid-January to early February. Fig.

6 shows these periods with light gray rectangles.

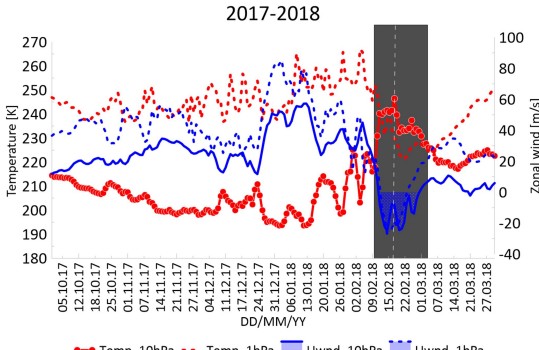

**Figure 5: Same as Figure 2, but Oct 2017 through Mar 2018.**

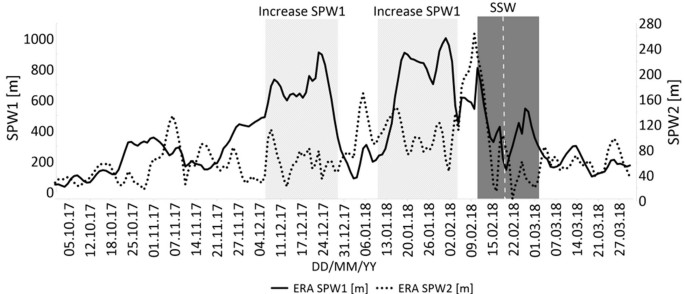

**Figure 6: Same as Figure 3, but Oct 2017 through Mar 2018.**



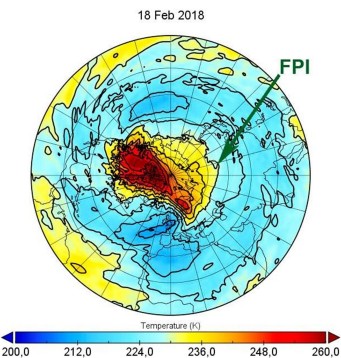


**Figure 7: Same as Figure 4.**

In the 2018-2019 winter, one SSW case was observed. The SSW emerged on 22 Dec and lasted until 19 Jan. The maximal temperature during that warming was 248K on 29 Dec (Fig. 8). A temperature increase was observed throughout the stratosphere. During the warming, the wind changed its direction to the westward. The FPI was within the area of warming

in that winter (Fig. 8). In Nov and Dec 2018, increased SPWs were observed; we marked those periods with light gray rectangles in Fig. 10. The latter shows that the warming covered the entire polar region, with the interferometer site being influenced by warm air in the stratosphere.

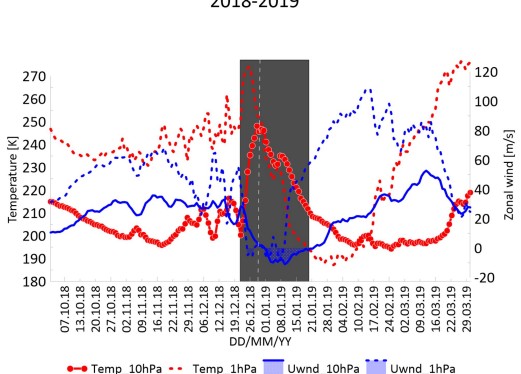

**Figure 8: Same as Figure 2, but Oct 2018 through Mar 2019.**



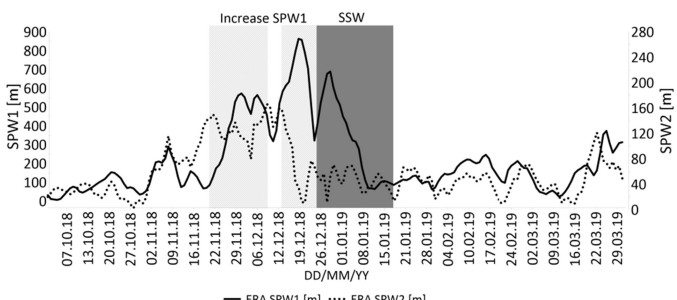

**Figure 9: Same as Figure 3, but Oct 2018 through Mar 2019.**

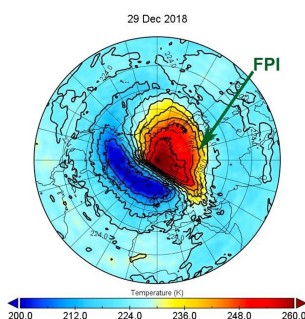

**Figure 10: Same as Figure 4.**

In the 2019-2020 winter, there were two atypical stratospheric warming cases shifted to a later date. The first warming was minor and lasted 30 Jan through 20 Feb with a maximal temperature of 239K on 5 Feb. SSW2 caused a significant temperature increase at 10 and 1 hPa; at 1 hPa, the zonal wind reversed. The SSW2 lasted 9 Mar through 28 Mar, the maximal temperature was 255K on 23 Mar. Two SPW increases preceded the SSW evolution in the stratosphere; we note that planetary waves were maximal for 2016-2020 (Fig. 12). During both SSW events, the TOR was within the warming

area. (Fig. 13)





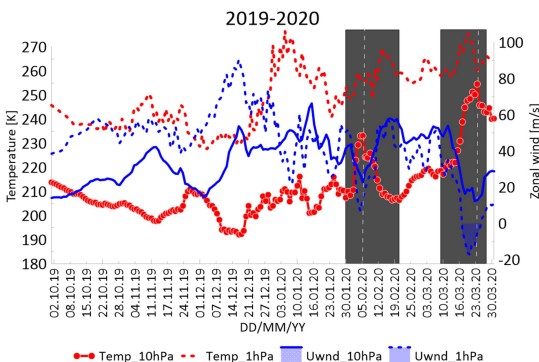

**Figure 11: Same as Figure 2, but Oct 2019 through Mar 2020.**

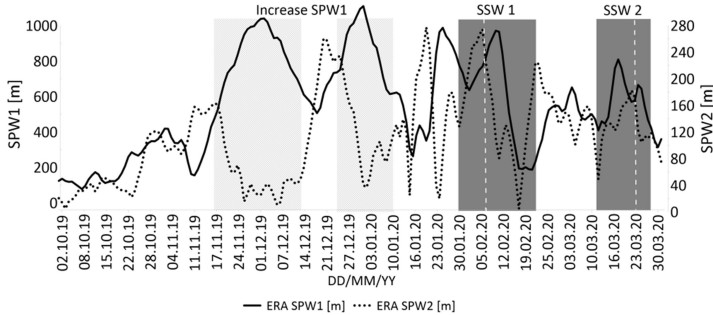

**Figure 12: Same as Figure 3, but Oct 2019 through Mar 2020.**

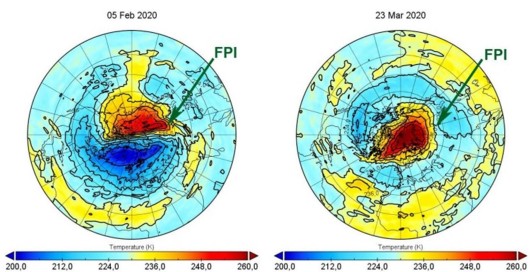

**Figure 13: Same as Figure 4.**



### 3.2 FPI-measured average night values of the 557.7-nm emission and temperature

In this section, we address the FPI-measured mean night values for the 557.7-nm emission, temperature, and zonal wind. In Figs. 14-15, we noted the SSW periods (grey rectangle), the day of maximal SSW temperature (dotted white vertical line), and the periods of the planetary wave increased activity (light grey rectangle). The amplitudes of stationary planetary waves were calculated along 60N with zonal wave numbers 1 and 2 (SPW1 and SPW2, respectively) from the Era5 data.

Fig. 14 shows the 557.7-nm emission decrease during the SSW. We can also see that an intensity decrease was observed during the periods without stratospheric warming. However, we note that low emission is always observed during the increased activity of planetary waves, especially SPW1. The MLT temperature is inversely related to the 557-nm emission. During SSW periods and increased activity of planetary waves, the temperature rises. The temperature rises may be explained by different heights of the radiation formation. The green line can be radiated from the heights with higher temperatures and reach values of up to 250 K, because the temperature height gradient over the mesopause can be extremely high (up to 10K/km). Analyzing the temperature behavior obtained with the 557.7-nm line, we can conclude that the green line emission shifted to the beginning of the thermosphere and decreased due to the inverse temperature dependence of the Barth mechanism (Barth, 1961). Thus, in addition to SSW, the planetary wave activity impacts on the MLT dynamics. The SPW activity most often precedes SSWs, but it may appear long before the warming onset and cause a response (often stronger than SSW) in the MLT temperature regime.

Preliminary analysis of the full vector of the wind velocity showed that obvious responses to the SSW and SPW events exist only for the zonal wind. However, this does not mean the absence of response from both the vertical and meridional wind on the lower atmosphere dynamics. We think that this should be investigated in a separate study. In this section, we address only the strongest and most obvious response of the zonal wind to SSWs and SPWs, obtained during the data analysis. The zonal wind shows a pronounced change during warmings. Like in the stratosphere, the eastward wind reverses westward in the MLT. Moreover, the stronger the wind inversion in the MLT, the stronger the wind inversion in the stratosphere. To analyze the effect of the stratosphere on the MLT wind, it is important to consider not only the standard 10 hPa height for SSW, but also the 1-hPa dynamics. For example, two 2017 SSW cases were minor as per the WMO classification. However, during these warmings at 1 hPa, the westward wind intensified significantly. This was the reason for the wind inversion at the MLT heights. The 2018-2019 warming was major, but at 1 hPa, there was no wind inversion. Also, we see that the zonal wind only weakened and did not reverse at the MLT altitudes.

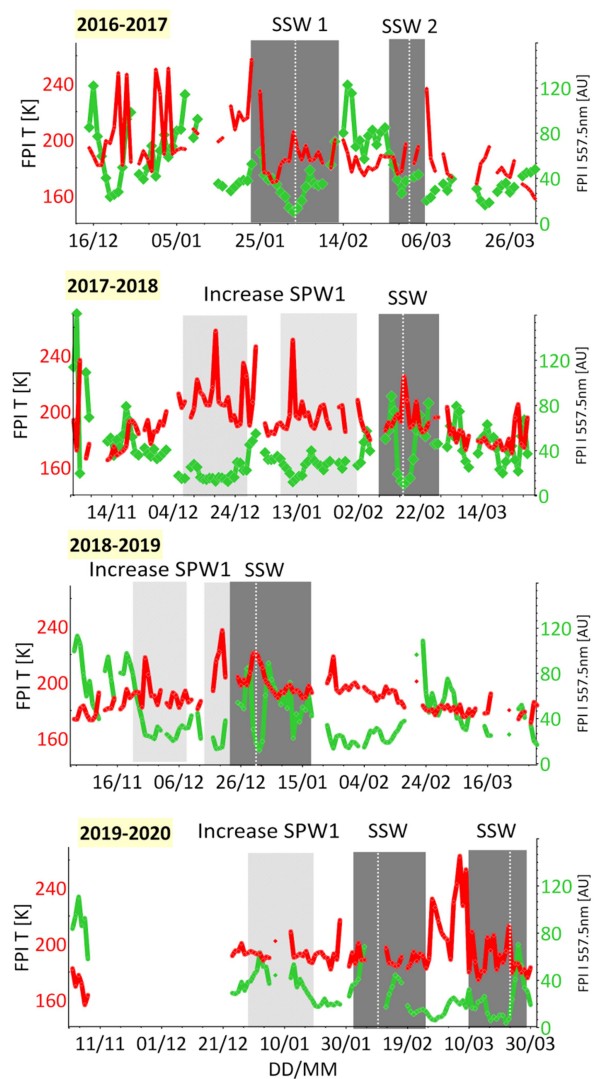

190

**Figure 14: FPI-measured 557.7-nm emission (green line), temperature (red line), the SPW1 amplitude increase period (light grey rectangle), the SSW periods (grey rectangle), and the day of the SSW maximal temperature (dotted white vertical line).**





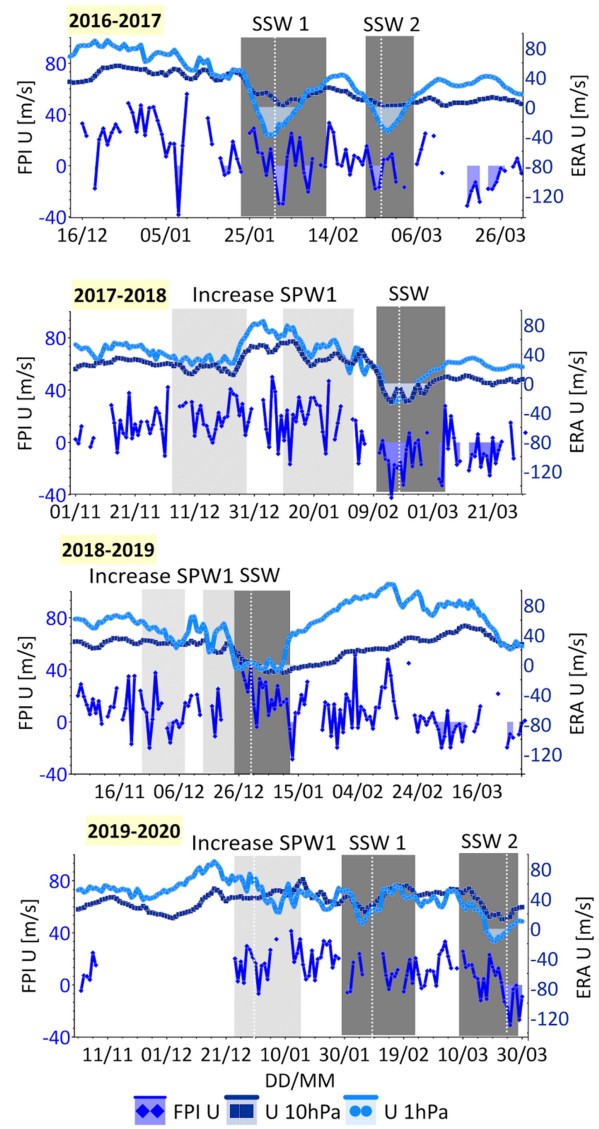

**Figure 15: FPI-measured 557.7-nm emission (green line), temperature (red line), the SPW1 amplitude increase period (light grey rectangle), the SSW periods (grey rectangle), and the day of the SSW maximal temperature (dotted white vertical line).**



### 3.3 Spectral analysis through the Lomb-Scargle method

Researchers also focused on diurnal variabilities of atmospheric characteristics during SSW impact (Merzlyakov et al., 2020; Pochotelov et al., 2018, Manson et al., 2002). They report the tide variability due to non-linear interactions with the planetary waves. The Lomb-Scargle (LS) periodogram method seems to be an appropriate technique for spectral analysis of the non-equidistant time rows, especially for FPI, because of gaps due to daytime, intense moonlight nights, and cloud covers. The Fig. 16 upper panel presents the calculated LS periodograms for the zonal wind variations observed during 2017-2018 winter. The main spectral components with 24-, 12-, 8-hour periods dominate in the upper atmosphere. Namely, these spectral components are present on the spectral characteristic for the zonal wind data. 12-hr oscillations have the largest amplitude. To check the validity of the retrieved spectral data, we prepared a testing sample of the data on the regular non-interrupted grid with 8-, 12-, and 24-hr spectral components having the 0.3, 1, and 0.3 amplitudes, respectively. The sample time step was 15 minutes, which corresponds to the FPI data minimal time step. We also added the normal noise to the data with zero mean and sigma equal to 3. The Fig. 16 bottom panel presents the LS periodogram for the artificial data sample. The Fig. 16 middle panel contains the LS periodogram of the described artificial data sample, but with the same gaps (daytime, moonlight, clouds) of the observed zonal wind during 2017-2018 winter. One can see a significant distortion in the spectral picture apparently due to the regular (daytime) 12-hr gaps that significantly increase the initial 8-hr and 24-hr spectral components and also generate additional spectral components. Therefore, a detailed spectral analysis for such non-regular data as the FPI samples is apparently impossible without additional information or some special processing of the initial datasets, or without significand modification of the analysis method. Still, we can estimate the diurnal variability of all tides by calculating the standard deviation of the diurnal dataset.





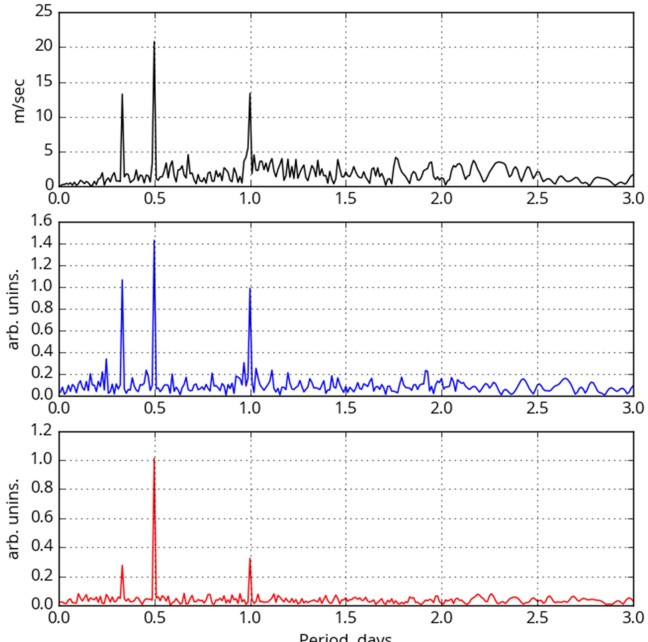

215

**Figure 16: LS periodogram of zonal wind (U) from the 2017-2018 FPI measurements.**

**3.4 Standard deviations of the FPI-measured 557.7-nm emission and temperature**

In this section, we analyze the 557.7-nm emission standard deviation, temperature, and zonal wind measurements for each night. During warmings and increased activity of planetary waves, emission fluctuations decrease, but temperature

220      fluctuations increase during night. Fig. 17 shows that temperature variations reach tens of degrees. Airglow and its night variations in the MLT were minimal during the periods of active stratosphere. Figs. 14 and 17 show that variations in the mean values for the emission and temperature correlate directly with the behavior of their standard deviations during the entire winter. This correlation is especially clear during SPWs and SSWs.

Unfortunately, in the 2019-2020 winter, there were technical problems with the interferometer measurements. Therefore,

225      some of the data are missing for that winter. However, during the 2019-2020 winter, we also see the opposite behavior of the mean values of temperature and emission, and their standard deviations. The only peculiarity of that winter is that a sharp temperature increase occurred several days before the NNE onset. While we cannot answer exactly, why this occurred, it is possible that the influence was exerted by the stratosphere dynamics at 1 hPa, because there were higher temperatures



throughout March at that height. In our opinion, the increase in temperature standard deviations is due to an increase in the

230    MLT tide amplitude, because the tides are the dominant mode in the MLT dynamics.

The variations in the wind standard deviation appear to be noisier, than those in the temperature standard deviation. Most often, we see that, during SSWs and SPWs, the zonal wind standard deviation increases. But this increase does not exceed the average background of wind variations even in the quiet stratosphere.

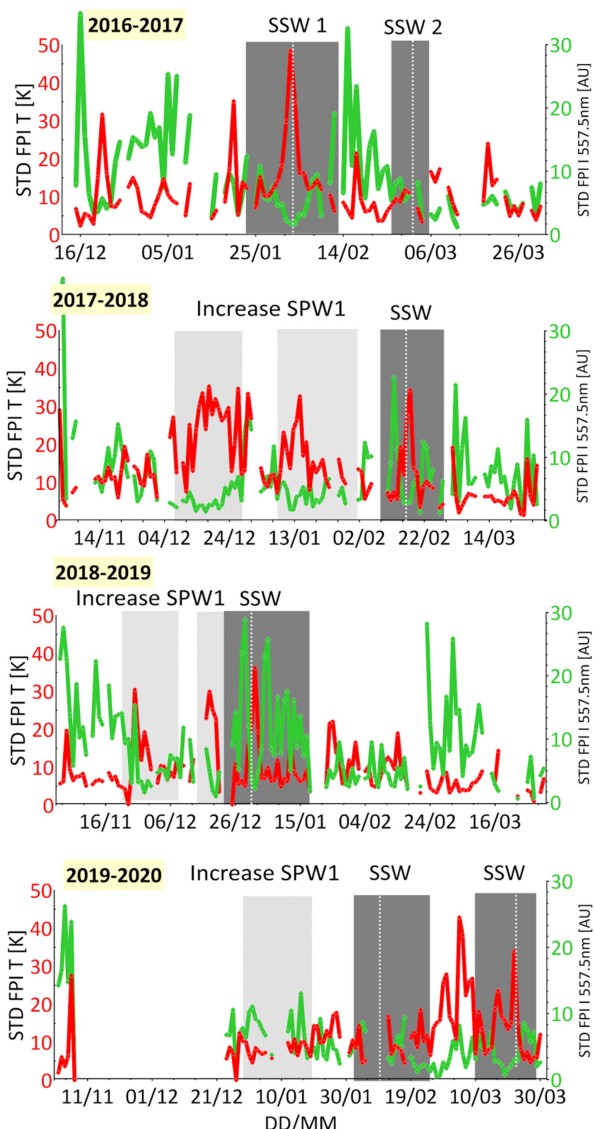

**Figure 17:** **FPI-measured standard deviation of the 557.7-nm emission (green line), of the temperature (red line), the period of the SPW1 amplitude increase (light grey rectangle), the SSW periods (grey rectangle), and the day of the SSW maximal temperature (dotted white vertical line).**



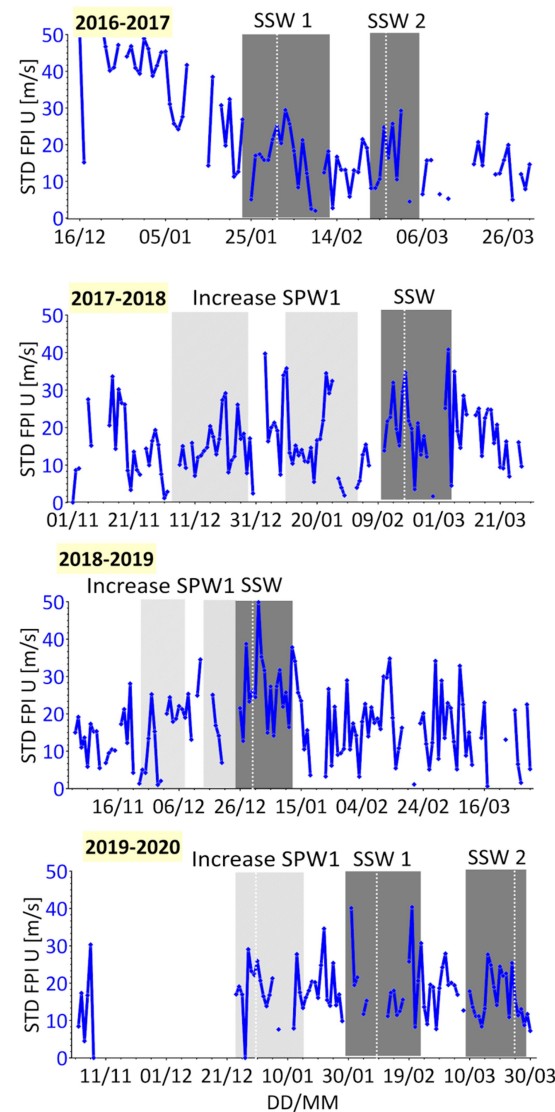

**Figure 18: FPI-measured standard deviation of zonal wind, the period of the SPW1 amplitude increase (light grey rectangle), the SSW periods (grey rectangle), and the day of the SSW maximal temperature (dotted white vertical line).**



## 4. Conclusion

We studied the behavior of the upper atmosphere during three winter seasons. We analyzed the mean values of wind and temperature, and their standard deviations during SSWs and SPWs in the stratosphere. Due to the nature of optical observations, we did not calculate tidal modes, but characterized the tides by dispersion for each nighttime measurement. The MLT response at the TOR appeared not to depend on the SSW location. The MLT responses equally, regardless of the SSW evolution over North America (2017-2018) or over the Asia-Pacific region (other winters). The 557.7-nm emission and its night variations decreased during SSWs and SPW activations.

The temperature and its variations rises sharply and significantly during the active stratosphere. Note that the response of the emission intensity and temperature is the same during the periods of SPW increased amplitude and during SSW events (sometimes, during active SPWs, it is even more significant). We suppose that, during SSWs and SPWs, a change in the height profile of the green emission layer in the atmosphere is possible. The lower part of the emission layer intensity near the mesopause may deplete, whereas the part of the emission layer over the mesopause persists unchanged or insignificantly increase due to a strong reverse temperature dependence of Barth mechanism reaction rate. Therefore, the FPI observes dimming of the integral emission intensity and corresponding increase in the tidal temperature variation, and the temperature itself during these events. A possible reason for observed effect is in the depletion of atomic oxygen for forming O (1S) state in the Barth mechanism near the mesopause due to the air vertical movement.

The response in the zonal wind is noticeable only during major SSWs. With a major SSW, the zonal wind reverses at MLT altitudes. The MLT wind inversion is observed during the wind inversion in the stratosphere; the height, at which this occurs does not matter. The zonal wind at the MLT heights does not respond to the dynamics of planetary waves. The zonal wind night fluctuations show no significant dependence on SSW/SPW activity. A possible reason may be weaker height gradients of the tidal amplitudes for the zonal wind as compared with the temperature, or more significant noise due to the data gaps. Hence, we cannot draw convincing conclusions about the tidal response of the wind during SSWs in this study.

*Acknowledgements*. The measurements were carried out on the instrument of Center for Common Use «Angara» [http://ckp-rf.ru/ckp/ 3056], the maintenance of the FPI was carried out with budgetary funding of Basic Research Program II.12. The study of the influence of stratospheric warming events to MLT was supported by the Russian Science Foundation, Project No. 19-77-00009. We also thank the European Center for Medium-Range Weather Forecast (ECMWF) for producing and making available their reanalysis ERA5 output.

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
