# Peer review of "Stratospheric influence on MLT over mid-latitudes in winter by"

_Annales Geophysicae, 2020_

## Referee Comment (RC1) · Anonymous Referee #1 · 14 Dec 2020

This paper presents a study on the influence of the stratospheric circulation on the mesosphere-lower thermosphere (MLT) based on observation of oxygen airglow emission (green line) in Eastern Siberia. The atmospheric layers are strongly coupled in winter when planetary waves can propagate upwards from the troposphere up to the MLT in the prevailing westerly winds. Observational evidences of this coupling are very welcome to improve our understanding of the processes involved. The most spectacular dynamical phenomena occurring in the middle atmosphere are the sudden stratospheric warmings (SSWs) in winter. They are always preceded by an amplification of planetary waves (wave 1 and wave 2). One originality of this paper is to consider not only the influence of SSWs on the MLT but also the influence of stationary planetary

waves. This document provides new information on the subject, but some additions and corrections are necessary to make it acceptable for publication in Annales Geophysicae as listed below:

- I do not understand why the term "stationary planetary wave" is used. The amplitude of the planetary wave (PW) is computed from ERA5 data from the given day but we do not know if the PW is stationary or travelling. The information of the phase evolution is needed to distinguish between stationary waves (constant longitude of the maximum) and travelling waves (longitude of the maximum shifting with time). It would be more appropriate to use only the term "planetary wave" or to show that the longitude of the PW is more or less constant.

- Observations are made on a middle latitude site (47.6N). The influence of PW phase should be considered. The TOR is at a longitude often close to position of the PW. To interpret the results I recommend to indicate also the spatial structure of SPWs as it is made for SSW (lines 113-115 and Figure 4). Is it the same for all SPW events ?

- Line 145. Please Indicate why you consider the 2019-2020 SSWs are atypical.

- Page 16, section 3.2., first paragraph. A more detailed description of the results is needed in this paragraph. It is said that the low emission is always observed during the increased activity of SPW1. This is not the case in 2019-2020 as well as for the increase of temperature. The temperature increase during SSWs does not occur always in the same part of the SSW period, at the beginning for SSW1 in 2016-2017, at the end for SSW 2 in 2016-2017, in the middle for SSWs in 2017-2018 and 2018-2019 and outside of the period for SSW 1 and 2 in 2019-2020.

- Line 183. It would be more logical to inverse the sentence: "the stronger the wind inversion in the stratosphere, the stronger the wind inversion in the MLT".

- Figure 16. Please add the signification of the three panels in the figure legend, not only in the text.
- Line 227. What is the NNE ? It is not defined.

- Section 3.4, lines 229-230. The increase in temperature standard deviation during SSW events is attributed to the increase in MLT tide amplitude. However the cause of this increase is not discussed in this section. Also it is not clear if the sentence refers only to winter 2019-2020 or to all winters. Looking at Figure 18, it seems that it is true also for other winters. An explanation is given in the conclusion where the increase in tide amplitude is attributed to the increase in the altitude of the emission layer during the SSW. This interpretation should be also discussed in section 3.4.

---

## Referee Comment (RC2) · Anonymous Referee #2 · 15 Dec 2020

In this paper, the wintertime atmosphere dynamics is analyzed, focusing on Sudden Stratospheric Warming (SSW) events. The data used were collected by the Fabry-65 Perot interferometer enabling the evaluation of the temperature and wind speed in the mesosphere-lower thermosphere (MLT) for four winter periods. These observations of the upper atmosphere have been compared with the corresponding measurements of the stratospheric dynamics obtained from the Era5 climate archive of the European Center for Medium-Range Weather Forecast (ECMWF). The results obtained are interesting and therefore the paper merits publication. However, there are weaknesses which can and must be removed, notably: 1. The work done on the extraordinary event of the first major Antarctic SSW which had as result the ozone hole split in Sep. 2002 has been ignored and must be cited. (https://link.springer.com/article/10.1007/BF02987584 ; https://link.springer.com/article/10.1007/BF02980093) 2. The Lomb-Scargle (LS) periodogram method used must be elaborated for the readers convenience, citing Lomb, N. R. 1976, Ap&SS, 39, 447 and Scargle, J. D. 1982, ApJ, 263, 835. 3. The use of the term Aństationary Âż planetary waves (SPWs is incompatible with the theory of the study and not accurate. 4. There are many spelling and grammatical errors in the text, and they need to be corrected.

In conclusion, I recommend publication after the above-mentioned revisions.

---

## Author Comment (AC1) · 15 Dec 2020

I and the co-author would like to thank the Referee for the comments. 1. Indeed, we have chosen the term "stationary planetary wave" incorrectly. We will replace this term with "planetary wave". Note that in the preliminary study, we considered the phase of the planetary wave, but did not find a significant relationship between the phase and processes in MLT. Most often, the wave is located to the east of our observatory. 2. We call the 2019-2020 SSW atypical because the warming started one month later than usual. We wanted to emphasize the atypically late time of the SSW occurrence. Perhaps we need to explain this more clearly in the paperwork. 3. I agree with the

referee, in section 3.2 we will discuss each SSW and PW case in more detail. Note that the periods of missing observations may contain not very correct data. As, for example, at the beginning of SSW1 in 2016-2017. Perhaps we will add a graph of the number of measurements during the day to each figure. And those days when there are few measurements we will consider with caution. I agree that there are questions with temperature variations, but oxygen emission always decreases with an active stratosphere. 4. Line 183. Absolutely. We'll fix it. 5. Figure 16. Ok, we'll add a legend. 6. Line 227. This is a typo, right - SSW. 7. Section 3.4, lines 229-230. Thanks for the recommendation, we will add interpretation to section 3.4.

---

## Author Comment (AC2) · 16 Dec 2020

We are very grateful to the Referee for comments. 1. Okay, we will read the papers (Varotsos, C. 2002; Varotsos, C. 2003) and we will cite them in our paperwork. 2. Thank you, this is a very valuable comment. We will add a citation to the paperwork (Lomb, N. R. 1976). 3. The first referee had a similar comment. We will use the term "planetary wave". 4. Unfortunately, I have an intermediate level of English and little experience in writing paper in English. I'm working hard on my English. I hope the corrector helps me fix all errors.

[Figure]

2020.

ANGEOD

---

## Author Response (AR1)

**RC1:**

**1)** I do not understand why the term "stationary planetary wave" is used. The amplitude of the planetary wave (PW) is computed from ERA5 data from the given day but we do not know if the PW is stationary or travelling. The information of the phase evolution is needed to distinguish between stationary waves (constant longitude of the maximum) and travelling waves (longitude of the maximum shifting with time). It would be more appropriate to use only the term "planetary wave" or to show that the longitude of the PW is more or less constant.

AC - Indeed, we have chosen the term "stationary planetary wave" incorrectly. We have replaced this term on "planetary wave" in the text and in figure captions (Fig 3,6,9,12,14,15,17,18).

**2)** Observations are made on a middle latitude site (47.6N). The influence of PW phase should be considered. The TOR is at a longitude often close to position of the PW. To interpret the results I recommend to indicate also the spatial structure of SPWs as it is made for SSW (lines 113-115 and Figure 4. Is it the same for all SPW events ?

AC – We reviewed 10 hPa synoptic charts for all winters. Usually, a stratospheric anticyclone occurs over the North Pacific and then it moves to the north of Canada. The temperature rises to the west of the anticyclone just in the observatory area, then the heat region shifts to the northeast. I give below pictures for days when there were high amplitudes of planetary waves. As you can see, the figures are similar. Perhaps, upon detailed analysis of the PW location, we will notice a difference. But in our opinion, the processes in the middle and upper atmosphere are global and we will register a response to PW and SSW, no matter where they developed - in the western or eastern hemisphere.

---

## Author Response (AR2)

Dear Theodore Giannaros.

We requested assistance from a native speaker Harding, Brian J. from University of California, Berkeley, United States (https://www.scopus.com/authid/detail.uri?authorId=55577884500, https://orcid.org/0000- 0002-1293-9379). Here is Brian's answer –

"I didn't think the English was too bad. Just a few spots needed some attention. I think the editor and / or reviewer were being too strict. It's a nice study. I'm not aware of previous FPI Some of the changes seem very large. I agree with your conclusion that it's difficult to interpret green line observations from the ground, since you can never be sure if you are looking at actual wind / temperature changes, or if you are looking at airglow height changes. "

We have corrected the text of the manuscript taking into account the corrections of the Brian Harding.

We also contacted the AnGeo editorial support team and received an answer - "We will do a light copyediting during the production process. You do not need to send it to a copy editing service".